# CeRNA Network Analysis Representing Characteristics of Different Tumor Environments Based on 1p/19q Codeletion in Oligodendrogliomas

**DOI:** 10.3390/cancers12092543

**Published:** 2020-09-07

**Authors:** Ju Won Ahn, YoungJoon Park, Su Jung Kang, So Jung Hwang, Kyung Gi Cho, JaeJoon Lim, KyuBum Kwack

**Affiliations:** 1Department of Biomedical Science, College of Life Science, CHA University, Seongnam 13488, Korea; eugene@chauniv.ac.kr (J.W.A.); yjparkep@chauniv.ac.kr (Y.P.); sujung_k@chauniv.ac.kr (S.J.K.); 2Department of Neurosurgery, Bundang CHA Medical Center, CHA University School of Medicine, Seongnam 13496, Korea; sjhwang7@chamc.co.kr (S.J.H.); sandori50@chamc.co.kr (K.G.C.)

**Keywords:** oligodendrogliomas, 1p/19q codeletion, long non-coding RNA, ceRNA network analysis, tumor microenvironment

## Abstract

**Simple Summary:**

Oligodendroglioma (OD) is a subtype of glioma occurring in the central nervous system. The 1p/19q codeletion is a prognostic marker of OD with an isocitrate dehydrogenase (IDH) mutation and is associated with a clinically favorable overall survival (OS). The long non-coding RNAs (lncRNAs) protects the mRNA from degradation by binding with the same miRNA by acting as a competitive endogenous RNA (ceRNA). Recently, although there is an increasing interest in lncRNAs on glioma studies, however, studies regarding their effects on OD and the 1p/19q codeletion remain limited. In our study, we performed in silico analyses using low-grade gliomas from datasets obtained from The Cancer Genome Atlas to investigate the effects of ceRNA with 1p/19q codeletion on ODs. We constructed 16 coding RNA–miRNA–lncRNA networks and the ceRNA network participated in ion channel activity, insulin secretion, and collagen network and extracellular matrix (ECM) changes. In conclusion, our results can provide insights into the possibility in the different tumor microenvironments and OS following 1p/19q codeletion through changes in the ceRNA network.

**Abstract:**

Oligodendroglioma (OD) is a subtype of glioma occurring in the central nervous system. The 1p/19q codeletion is a prognostic marker of OD with an isocitrate dehydrogenase (*IDH*) mutation and is associated with a clinically favorable overall survival (OS); however, the exact underlying mechanism remains unclear. Long non-coding RNAs (lncRNAs) have recently been suggested to regulate carcinogenesis and prognosis in cancer patients. Here, we performed in silico analyses using low-grade gliomas from datasets obtained from The Cancer Genome Atlas to investigate the effects of ceRNA with 1p/19q codeletion on ODs. Thus, we selected modules of differentially expressed genes that were closely related to 1p/19q codeletion traits using weighted gene co-expression network analysis and constructed 16 coding RNA–miRNA–lncRNA networks. The ceRNA network participated in ion channel activity, insulin secretion, and collagen network and extracellular matrix (ECM) changes. In conclusion, ceRNAs with a 1p/19q codeletion can create different tumor microenvironments via potassium ion channels and ECM composition changes; furthermore, differences in OS may occur. Moreover, if extrapolated to gliomas, our results can provide insights into the consequences of identical gene expression, indicating the possibility of tracking different biological processes in different subtypes of glioma.

## 1. Introduction

Gliomas are the most frequently occurring tumors in the central nervous system [1,2]. Oligodendrogliomas (OD), a subtype of gliomas, account for <10% of all primary gliomas [3,4]. In 2016, the World Health Organization (WHO) established a new classification system for central nervous system tumors by adding newer genetic molecular markers to existing histopathological characteristics. *IDH1* and *IDH2* (collectively referred to as *IDH* hereafter), which encode isocitrate dehydrogenase and cause mutations and 1p/19q codeletion, were included as characteristic prognostic markers of ODs [5]. Although the 1p/19q codeletion is a prognostic marker of ODs with *IDH1/2* mutations [6,7], 30–40% of all ODs are not codeleted at 1p/19q and have a worse prognosis [8,9,10]. Hence, the 1p/19q codeletion is reportedly associated with diagnosis, prognosis, and clinically favorable overall survival (OS) in glioma; however, the exact underlying mechanism remains unclear [3,11,12,13,14,15,16].

With the development of high-throughput sequencing, the genomic functions of approximately 2% of the human genome have been identified [17,18,19], including those of long non-coding RNAs (lncRNAs) with pseudogenes [20]. However, most lncRNAs have not been identified based on their biological importance [21]. lncRNAs are regulatory RNA transcripts with a length >200 nucleotides [22] and have recently been reported to play various important roles in the regulation of transcription and translation as well as in epigenetic modification [19,23].

The miRNAs play roles as inhibitory regulators at the post-transcriptional level by binding their specific target site known as miRNA response element (MRE) [24]. The lncRNAs share MRE sequences with a coding gene, which protects the mRNA from degradation by binding with the same miRNA and also by acting as a competitive endogenous RNA (ceRNA) [20,25]. The epigenetic change of their expression has been related with various diseases. To be more specific, several oncologic studies have reported that lncRNAs, particularly ceRNAs play important roles in tumorigenesis and tumor progression [26,27,28]. The first report of the ceRNA in cancer is phosphatase and tensin homolog pseudogene 1 (*PTENP1*), a pseudogene of phosphatase and tensin homolog (*PTEN*), and it showed tumor suppressive effect by binding miRNAs such as miR-17, miR-19, and miR-21 which were ultimately targeting *PTEN* [29]. The tumor suppressive effects of *PTEN* in the ceRNA network have also been reported in conjunction with growth arrest-specific 5 (*GAS5)*, Itchy E3 Ubiquitin Protein Ligase (*Circ-ITCH*), Fer-1-like protein 4 (*FER1L4*) [30,31,32]. On contrary, some noticeable studies also reported on the oncogenic effect of lncRNA such as KRAS proto-oncogene, GTPase pseudogene 1 (*KRASP1*), and H19 imprinted maternally expressed transcript (*H19*). *KRASP1* affects tumor growth and development by competitive binding for shared miRNAs with KRAS proto-oncogene, GTPase (*KRAS*) [29]. *H19* is mutually counter-regulated by tumor suppressor p53 [33].

In a glioma study, they report on the oncogenic role of maternally expressed 3 (*MEG3*) as a tumor suppressor lncRNA that is downregulated in gliomas [34] and nuclear paraspeckle assembly transcript 1 (*NEAT1*) on by competing for miR-let-7e and modulating the expression of signal transducer and activator of transcription 3 (*STAT3*) and NRAS proto-oncogene, GTPase (*NRAS*) [34,35]. The functions of lncRNAs as biomarkers and therapeutic targets in glioma are well reported [36,37,38]; Collectively, these findings showed that the interactions among lncRNAs, mRNA, and miRNA in ceRNA network implicate a potential role as tumor suppressive or oncogenic effects. However, data regarding their effects on OD and the 1p/19q codeletion remain limited. Hence, in this study, we identified 16 lncRNA–miRNA–mRNA networks in ODs with 1p/19q codeletion using The Cancer Genome Atlas (TCGA) database. 

## 2. Materials and Methods

### 2.1. Processing Genomic and Clinical Profiles from TCGA Database

Gene expression quantification data of mRNA, lncRNA, pseudogene, and miRNA isoforms of low-grade gliomas (LGG), inclusive of OD, and the corresponding clinical profiles were downloaded, normalized, and processed within the TCGAbiolinks pipeline (version 2.15.3) in R software (version 3.5.2) [39].

We downloaded harmonized data from TCGA database as reference of hg38 (platform “Illumina HiSeq”) for 189 patients with OD who had IDH mutation information and selected 168 patients with IDH mutation. The function GDCprepare used with the options legacy = FALSE, datacategory = “Transcriptome Profiling”, datatype = “Gene Expression Quantification”, workflowtype = “HTSeq − Counts” for obtaining mRNA transcripts data. The function GDCprepare used with the options legacy = FALSE, datacategory = “Transcriptome Profiling”, datatype = “Isoform Expression Quantification” and workflowtype = “BCGSC miRNA Profiling” for obtaining miRNA transcripts data (reads per million mapped reads, RPM).

Then, we performed the pre-processing steps according to the TCGA’s workflow and standard pipeline as recommended in TCGAbiolinks [39,40]. We checked to possible outliers for mRNA transcript data using the TCGAanalyze_Preprocessing function with the options cor.cut = 0.6 (Correlation coefficient). Then, we normalized mRNA transcripts data using the TCGAanalyze_Normalization function to adjust for the GC contents effect and followed gene filtering steps using the TCGAanalyze_Filtering functions. The miRNA transcripts data was filtered by quantile using the TCGAanalyze_Filtering functions, and lowly expressed miRNA with less than overall average RPM 3 was filtered out and transformed into log2(RPM + 1). Moreover, all genes of both mRNA and miRNA for which the expression level in one patient was greater than one-third of the sum of total patients’ gene expression levels were excluded.

To classify non-coding RNAs, protein-coding gene, long non-coding gene, and pseudogene datasets were downloaded from the HUGO Gene Nomenclature Committee (HGNC) database. Furthermore, the data from the HGNC dataset were used to identify Ensembl IDs. miRBase accession numbers were converted to miRNA symbols using miRBaseConvertor (version 1.10.0), and miRNA expression levels were re-quantified for every miRNA symbol [41].

### 2.2. Analysis of Differentially Expressed Genes and Their Visualization

Differentially expressed genes (DEGs), including coding RNAs (DEcodingRNAs), non-coding RNAs (DEncRNAs) such as lncRNAs (DElncRNAs), pseudogenes (DEpseudogenes), and miRNAs (DEmiRNAs), were identified using the edgeR package provided in the TCGAbiolinks differential expression analysis (DEA) process. The filter criteria were |Log2 fold change| > 1 and *p* value of <0.05 using false discovery rate (FDR) correction; results were presented via heatmaps generated using the TCGAbiolinks package [32]. For better identification and analysis in terms of the distribution and chromosomal locations of DEGs, ShinyGO [42], and Genome Decoration Page (GDP, source: http://www.ncbi.nlm.nih.gov/genome/tools/gdp) [43] were used for visualizing the data. ShinyGO serves as a statistical analysis tool and identifies gene characteristics such as genotypes, GC content distribution, and chromosomal locations between query genes and background data. GDP is a visualization tool which shows distribution of query genes on the genome.

### 2.3. Weighted Gene Co-Expression Network Analysis

We calculated the similarity matrix (*S_ij_*) of expression between DEcodingRNAs and DE-ncRNAs using Pearson’s correlation coefficients. Next, for measuring contributions among genes and to transform the similarity matrix into an adjacency matrix (*a_ij_*), the following formula was used:(1)$$a_{ij}={\left|0.5\;\times \left(1+cor\left(i,j\right)\right)\right|}^{\beta }$$

The soft threshold statistical powers (*β*) of upregulated DEcodingRNAs and DEncRNAs and downregulated DEcodingRNAs and DEncRNAswere both set to 4. Accordingly, scale-free topology fit index and mean interconnectivity were measured and topological overlap matrix (TOM) was performed. Next, topological overlap dissimilarity measure (dissTOM) was assessed, and using the obtained value, a hierarchical clustering tree was constructed using the average hierarchical clustering method to group the DEcodingRNAs and DEncRNAs. A dynamic cut tree algorithm with minimum a module size of 15 and merging threshold over similarity of 0.7 (cutoff of 0.3) was used to generate the appropriate number of and densely interconnected modules between DEcodingRNAs and DEncRNAs. Next, the function of module eigengenes (MEs) was found to be represented as the first principal component of the expression matrix, and we could identify highly correlated modules with clinical traits using Pearson’s correlation tests using MEs in WGCNA [44]. We evaluated the profile of each module by comparing it to the correlation between the modules and the clinical traits of OD having an IDH mutation with a 1p/19q codeletion.

### 2.4. Construction of a CeRNA Network

We performed DEA and WGCNA to select candidate genes for the ceRNA network and obtained positively co-expressed DEcoding RNAs and DEncRNAs and negatively expressed DEmiRNAs in OD patients with *IDH* mutations and 1p/19q codeletions. Next, the following bioinformatics tools were used to predict the interactions among the selected DEcodingRNAs, DEncRNAs, and DEmiRNAs. RNAInter (https://www.rna-society.org/rnainter/) [45] is a platform containing comprehensive RNA-associated interactomes utilizing 35 integrated computationally predicted and experimentally validated databases, including TargetScan, miRcode, and miRanda. ENCORI (http://starbase.sysu.edu.cn/index.php; version 3.0) [46] and mirDIP 4.1 (http://ophid.utoronto.ca/mirDIP/) [47] provide integrated data of miRNA–target interactions from across diverse resources. Basically, we acquired both the prediction of DEncRNA–DEmiRNA interactions and DEcodingRNA–DEmiRNA interactions using these databases. Additionally, we used other databases for supplementary and validation purposes. DIANA-TarBase v8 (www.microrna.gr/tarbase) [48] and miRWalk (http://mirwalk.umm.uni-heidelberg.de/; version 3.0) [49] were used for DEcodingRNA–DEmiRNA interactions and the DIANA-LncBase v2 was used for DEncRNA–DEmiRNA interactions.

### 2.5. Evaluation and Visualization of CeRNA Network

We evaluated the ceRNA network using adjusted rand index (ARI) with DEcodingRNAs belonging to each ceRNA network and selected it as the final ceRNA network having a high concordance with traits of *IDH* mutations with 1p/19q codeletion. Finally, the ceRNA network with an ARI of over 0.6 was visualized using Cytoscape software 3.7.2 [50].

### 2.6. Gene Ontology and Pathway Enrichment Analysis

Gene ontology (GO) analyses—including those for biological processes, cellular components, and molecular functions—were performed using ShinyGO with an FDR of <0.05 [42]. The Kyoto Encyclopedia of Genes and Genomes (KEGG) and reactome pathway enrichment analyses were performed using ClueGO software with a Bonferroni step-down of <0.05 in Cytoscape [51]. The results of the enrichment analyses were visualized using ShinyGO and Cytoscape software.

### 2.7. Statistical Analysis for Validation of the CeRNA Network

Agglomerative unsupervised clustering using the ward method was performed for validating the ceRNA network, with an expression pattern of gene sets of DEcodingRNA belonging to each ceRNA network, and the results were compared with actual clinical data of patient subgroups using Python software (version 2.7; Python Software Foundation, Wilmington, DE, USA). The number of clusters (set on unsupervised clustering) was 2 for OD and 3 for low-grade glioma patients. We also compared the survival rate between the actual subgroup and clusters from unsupervised clustering using the Kaplan–Meier survival analysis. Furthermore, two-dimensional K-means clustering and survival rate were assessed to identify significant clinical effects of lncRNAs as ceRNA in R (version 3.5.2; R Foundation, Vienna, Austria) software and GEPIA [52].

## 3. Results

### 3.1. Differentially Expressed Gene Analysis of mRNAs, miRNAs, and ncRNAs

TCGA is the world’s largest cancer database and is supported by the National Cancer Institute (NCI) and the National Human Genome Research Institute (NHGRI). TCGA is a bounty of multi-omics data and clinical information, including genetic information for various cancers. In this study, the expression data of codingRNAs, ncRNAs, and miRNAs of a total of 168 OD patients from the TCGA lower grade glioma (LGG) database were analyzed. Most of these patients had *IDH* mutations, but only 127 patients had 1p/19q codeletions. We also performed differential expression analysis of codingRNAs, lncRNAs, and miRNAs according to fold change values and significance levels to identify the quantitative differences among genes. The distribution of DEcodingRNAs (Figure 1), DE lncRNAs, DEpseudogenes (Appendix A), and DEmiRNAs (Figure 2) was directly visualized using volcano plots and ideogram. There were 342 upregulated DEcodingRNAs and 573 downregulated DEcodingRNAs in *IDH* mutations with 1p/19q codeletions than in those with non-codeletions (Figure 1A). Next, we confirmed the types and positions of genes using ShinyGO. The types of all inputted DEcodingRNAs were codingRNAs, and downregulated codingRNAs were significantly distributed on chromosome 1 and 19 compared to the background data (Figure 1B). Although there was no statistically significant difference from the background data, many of the upregulated codingRNAs were relatively distributed on chromosome 17. Furthermore, 38 DElncRNAs and 7 DEpsuedogenes were upregulated, and many of these genes were relatively upregulated on chromosomes 6 and 17. Moreover, 42 DElncRNAs and 5 DEpseudogenes were downregulated and distributed on chromosomes 1 and 7 (Appendix A). We also detected 3 upregulated DEmiRNAs and 29 downregulated DEmiRNAs (Figure 2A). The upregulated miRNAs were distributed on chromosomes 6 and X. Many of the downregulated miRNAs were relatively distributed on chromosomes 1, 8, 9, 17, and X (Figure 2B). The upregulated DEmiRNAs of chromosome X (hsa-miR-888 and has-miR-891a) were located very closely together, therefore indicated by a single arrow in the Figure 2B. Downregulated DEmiRNAs which have 5p/3p pairs from same family (i.e., hsa-miR-204-5p/3p on chromosome 9, hsa-miR-301a-5p/3p, hsa-miR-3065-5p/3p on chromosome 17 and hsa-miR-1298-5p/3p, hsa-miR-2114-5p/3p on chromosome X) are also presented with single arrows in Figure 2B.

### 3.2. Analysis of a Weighted Co-Expression Network and Identification of Key Modules

For ncRNAs such as lncRNAs and pseudogenes to function as ceRNAs, coding RNAs within the ceRNA network is a potential regulatory target of ncRNAs, and the gene expressions of ncRNAs and coding RNAs are positively correlated. Weighted gene coexpression network analysis (WGCNA) can cluster highly correlated genes between DEcodingRNAs and DEncRNAs into the same module and identify significant candidate modules related to the clinical traits of cancer. We obtained a total of 915 DEcodingRNAs (342 upregulated and 573 downregulated) and 92 DEncRNAs (45 upregulated and 47 downregulated) from previous DEAs for constructing the coexpression network via WGCNA. In the first step of WGCNA, the soft-threshold power (β) from 1 to 20 was calculated considering scale independence and mean connectivity and was confirmed to be 4 in both network topology with upregulated and downregulated genes (Figure 3A). Next, we set 15 as the minimum module size and identified co-expression modules between DEcodingRNAs and DEncRNAs using dynamic cut tree. We also calculated the similarity and dissimilarity among modules to merge modules having extremely similar expression profiles. MEDissThres, the threshold for dissimilarity, was set as 0.3 and modules having high correlation >0.7 were merged into one (Appendix A). We finally obtained 6 modules each in the upregulated and downregulated co-expression network (Figure 3B) and assessed which modules correspond more with the clinical traits by evaluating the associations between the modules and clinical traits (Figure 3C). The results were that the blue module had the strongest positive relationship with the 1p/19q codeletion of *IDH* mutation in ODs, with a correlation value of 0.69 (left panel of Figure 3C); the turquoise module had the strongest negative relationship, with a value of −0.71 (right panel of Figure 3C).

### 3.3. Construction of CeRNA Network and Prediction of Function in Oligodendroglioma

According to the ceRNA theory, ncRNAs competitively bind to miRNA response elements (MREs) of miRNA and act as positive regulators for the regulation of mRNA activity. Thus, we built 26 ceRNA networks such that DEmiRNAs matched with gene sets from the blue and turquoise modules. The target prediction of DEmiRNA was achieved using RNAInter, ENCORI (starBase v3.0), mirDIP, miRWalk, LncBase, and Tarbase. In the blue module, 24 downregulated DEmiRNAs interacted with 218 upregulated DEcodingRNAs, 24 upregulated DElncRNAs, and 4 upregulated DEpseudogenes. Furthermore, 2 upregulated DEmiRNAs were matched with 290 downregulated DEcodingRNAs, 6 downregulated DElncRNAs, and 2 downregulated DEpseudogenes from the turquoise module.

Next, we evaluated the ceRNA networks to select networks that better reflect clinical characteristics. Adjusted rand index (ARI) has been used as criterion to measure the similarity between two data partitions. An ARI value close to 1 indicates an almost perfect agreement, whereas a value of 0 indicates discordance between the partitions. Thus, we evaluated each gene set of DEcodingRNAs involved in 26 ceRNA networks using ARI; finally, 16 ceRNA networks having with an ARI of >0.6 remained from just the blue module, and these networks included 16 downregulated miRNAs, 209 upregulated codingRNAs, 24 lncRNAs, and 4 pseudogenes (Appendix A). We visualized these ceRNA networks using Cytoscape version 3.7.2 software (Figure 4A).

Subsequently, we performed Gene Ontology (GO) and KEGG analyses and the reactome pathway enrichment analysis using ShinyGo and ClueGO to clarify the potential functions of the ceRNA networks that may be involved in tumorigenesis and carcinogenesis (Figure 4B,C and Appendix A).

The genes related to the ceRNA network were significantly enriched in nervous system development, cell-to-cell signaling, cell differentiation, neuron projection terminus, collagen network, MAP kinase phosphatase activity, DNA-binding transcription activator activity, and RNA polymerase II-specific processes (Figure 4B). Furthermore, enriched pathway terms were mainly associated with ECM–receptor interactions, RAF-independent MAPK1/3 activation, collagen biosynthesis and modifying enzymes, prostate cancer, platelet homeostasis, NCAM1 interactions, and signaling by NTRKs (Figure 4C). The results of the functional enrichment analyses suggested that these genes from the ceRNA network could participate in the tumorigenesis and development of OD with 1p/19q codeletion by regulating the related biological processes and key pathways.

### 3.4. Extending the CeRNA Network to Lower Grade Gliomas and Verifying the Clinical Significance through Survival Analysis

Additionally, we assessed the ceRNA network using ARI in low glioma gliomas to determine the potential effects of ceRNA networks in other types of glioma cancers. The evaluation was based on 1p/19q codeletion, 1p/19q non-codeletion, and wildtype, and the results surprisingly confirmed that all ceRNA networks distinguish well between the 1p/19q codeletion and wildtype in lower grade gliomas (Table 1). This result indicates that the ceRNA network, according to the 1p/19q codeletion, distinguishes each subgroup well not only in OD but also in the entire lower grade glioma and that it has characteristics that can represent the biological differences among subgroups.

Next, we performed the OS analysis to confirm ceRNAs of clinical importance according to their expression levels in the ceRNA network. As the clinically important ceRNAs were expected to exhibit a strong negative relationship with target miRNAs, expected ceRNA levels were assessed for the correlation coefficient between miRNAs in each ceRNA network, and a value of less than −0.3 was selected. Subsequently, k-means were used to classify groups with high and low expression levels, and the OS of each group of patients in OD and lower grade glioma was compared (Table 2).

As a result, patients with *IDH* mutations in OD had different survivals according to the expression levels of lncRNA *INHBA-AS1* and *LINC01551*. This is a clinically meaningful result considering that the 1p/19q codeletion group and the non-codeletion group were not statistically significantly divided in terms of the data of patients with OD with *IDH* mutation. In addition, in the data of patients extended to lower grade glioma, a higher number of non-coding RNAs in the ceRNA network showed a difference in OS rate according to the expression level. In particular, the ceRNA network containing *LINC01551* and hsa-miR-301a-3p similarly classified subgroups of 1p/19q from both OD and lower glioma including wildtype compared to the actual subgroup (Figure 5A,B and Appendix A). In addition, the survival rate according to the expression level of *LINC01551* was also significantly different in both OD and lower grade glioma patients (Figure 5C and Appendix A).

## 4. Discussion

Based on the WHO classification and various research reports, the 1p/19q codeletion is used as a prognostic biomarker [53]. Some parts of genes related to the 1p/19q codeletion can promote tumor progression, but in many cases, this is a positive prognostic indicator not only for OD but also for glioma [8,11]. Recently, the functional studies on ncRNA, which are expected to play a role in ceRNA, are being conducted, and important roles of ncRNAs in various biological processes have been reported in studies related to tumorigenesis and metastasis [26,27,28]. In studies on gliomas, various studies on ncRNAs that are expected to play a role as ceRNAs, including lncRNA, are in progress, but no studies on 1p/19q have been reported. Therefore, a better understanding of the effect of 1p/19q codeletion on OD will be of great help to an improved understanding of treatment and cancer progression in OD as well as overall glioma.

Based on the ceRNA hypothesis, a total of 16 mRNA–lncRNA–miRNA networks reflecting 1p/19q codeletion characteristics were constructed, and 16 miRNAs, 209 mRNAs, and 28 ncRNAs were mapped in our study. The ceRNA network we discovered was evaluated using ARI, and the network is very similar to the classification of the actual subgroup indicated in the clinical data of the actual TCGA data and reflects its characteristics well (Appendix A).

We performed the GO and KEGG/reactome pathway analyses to identify the biological processes and molecular functions of ncRNA in the ceRNA network and revealed features that can play a pivotal role in tumorigenesis (Figure 4B,C).

The results of the GO and pathway enrichment analyses were enriched in terms related to ion channel activity, insulin secretion, and MAP kinase phosphatase activity as well as extracellular matrix (ECM)-related terminologies such as ECM degradation, collagen network, etc.

Potassium ion channels are known as cancer hallmarks and play roles in the evasion of apoptosis, proliferation, and invasion by their potential to alter membrane polarization [54,55,56]. Two-pore domain (K_2P_) potassium channels enable the resting currents of potassium ions. *KCNK1*, *KCNK3*, and *KCNK10* are substrate genes of the two-pore-domain (K_2P_) potassium channel family and are differentially distributed in the CNS and are known to be involved in neuronal functional maintenance and modulation [57,58]. In a related study, *KCNK1*, *KCNK3*, and *KCNK10* were significantly downregulated in brain glioblastoma (GBM) [59]. In particular, the expression of *KCNK1* was decreased more in cancers of the overall CNS—such as astrocytomas (AC), GBMs, ODs, medulloblastomas, and melanomas—than in the normal tissue. Additionally, ATP-sensitive potassium channels, which comprise pore-forming subunit (Kir6.2) and regulatory sulfonylurea receptors (SUR1), are known to control the glucose-stimulated release of insulin. In our research, several genes, such as *ABCC8* and *KCNJ11*, related to insulin and diabetes and encoding Kir6.2 and SUR1 have also been found in the ceRNA network.

The repression of Kir6.2/SUR1 channels stimulates the release of insulin [60], and it has been reported that mutations and deficiencies of *ABCC8* and *KCNJ11* induce hyperinsulinism [61,62,63,64,65]. In various studies and clinical reports, the hypersecretion of insulin has been linked to the development of several types of cancer, including breast, colon, liver, and kidney cancers [66,67], and the survival rates through GEPIA in our research also confirmed that higher the expression levels of the two genes, the better is the survival rate (Appendix A). These genes were relatively upregulated as per the results of our study, and these results suggest that potassium channels play important roles in the brain tumor microenvironments related to proliferation, apoptosis, and migration; this can be one of the survival advantages of the 1p/19q codeletion in gliomas.

The ECM is composed of diverse biochemical components including collagens and glycoproteins such as tenascins. The tumor microenvironment, including the ECM, is subject to changes in various biological conditions—including the structure, content, and distribution—during most of the cancer development duration and progression, unlike that in normal cells [68]. Tenascins are a family of multioligomeric glycoproteins that are primarily expressed in the CNS, and these are known as one of the ECM components participating in cell adhesion, migration, development, and proliferation. Tenascins normally exist in the hexameric form, such as tenascin-C (TNC), tenascin-X (TNX), and tenascin-W (TNW), or in the trimeric form, such as tenascin-R (TNR). TNC is overexpressed in malignant gliomas such as high grade astrocytomas and glioblastomas and promotes the invasion of malignant gliomas [69,70,71]. In contrast, TNR is highly expressed in normal oligodendrocytes, neurons, and non-invasive gliomas such as pilocytic astrocytic tumors, but its expression decreases in malignant astrocytoma [72]. Interestingly, the high expression of *TNR* gene has been observed in ODs with a high grade in this report, and the expression of *TNR* is upregulated in OD having a 1p/19q codeletion. Thus, we performed survival analysis using GEPIA and confirmed that the higher the expression level of *TNR* the better is the survival rate in OD patients (Appendix A). Moreover, in a recent study, the expression of *TNR* was reported as a new prognostic marker for GBM using machine learning [73].

The invasion process of cancer involves the degradation of these ECM components by matrix metalloproteinases (MMP) as well [74]. *MMP24* is a gene encoding matrix metalloproteinase-24 and has recently been reported to negatively regulate the aggressiveness of cancer cells in breast cancer as a target gene of yes-associated protein (YAP) [75].

In a previous study on glioma, the expression of *MMP24* was high in brain tumors and was reportedly to related to tumor progression [76], but the available data for *MMP24* in later study seemed to be contradictory [77]. The interpretation of our findings on *MMP24* was also challenging. The expression of *MMP24* was observed to be high in GBM and OD, similar to that reported in previous studies, but patients with a high level of *MMP24* had better survival in OD (Appendix A). For these reasons, further studies on *MMP24* in glioma are needed.

In addition, *SLC8A2*, which is a member of the solute carrier family 8, is a tumor suppressor gene of GBM [78] and shows a difference in the survival rate, according to the expression level in OD of TCGA. *SLC8A3* from our findings also exhibited the same pattern in expression and survival rate as *SLC8A2* in OD of TCGA, but the related reports are still insufficient and further studies need to be conducted.

Kaplan–Meier survival analysis was performed to investigate the clinical characteristics in the ceRNA network we discovered. The k-means analysis was performed based on the gene set of the mRNA contained in the ceRNA to divide the unsupervised subgroups and then compare survival rates with actual subgroups. The results were confirmed to be similar to the survival rate according to the clinical information of actual TCGA OD patients with *IDH* mutations. Characteristically, the difference in survival rate among patients according to actual 1p/19q codeletion subtypes in the OD data of TCGA was not statistically significantly separated, but a difference of survival rate in the group as per the expression level of ceRNA was divided (Figure 5B). The results indicate that the expression pattern of the gene depending on the 1p/19q codeletion is an important factor that greatly influences the survival and prognosis of patients with glioma. In addition, the Kaplan–Meier survival analysis was performed by classifying the expression level of ncRNA compared to that of miRNA into high and low expression groups (Figure 5C), and we found clinically important ncRNAs within the ceRNA network (Table 2). Based on these results, we wanted to extend the characteristics identified in OD to lower grade glioma and confirm valid results. The excavated ceRNA networks classified the subgroups of lower grade glioma to the correct level (Appendix A) and showed a difference in the survival rate (Appendix A), which have applicable potential to demonstrate a fairly similar pattern and biological influence derived from the 1p/19q codeletion in the development and progression of glioma (Appendix A). Additionally, we analyzed the clinical key roles of expected ncRNAs in the ceRNA network and confirmed diverse potentials, which highlighted few limitations of this study (Table 2). Although a previous report on hepatocellular carcinoma indicated that *LINC01551* promotes tumorigenesis [79], the result of *LINC01551* belonging to the ceRNA network related to hsa-miR-301a-3p was clinically positive in our study. Furthermore, the comparison results of gene expression among the subtype of lower grade glioma through GEPIA, the expression amount of *LINC01551* in OD, was more similar to that of normal brain tissue than to that of other subtypes (Appendix A). hsa-miR-301a-3p, which was downregulated in our study, promotes the invasion of glioma, and these results are supported by the findings of our previous research [80,81]. *INHBA-AS1* had mostly negative reports in tumorigenesis [82], but the results were the same as those for *LINC01551* in our study with OD with *IDH* mutations (Appendix A).

In addition, in the results of an expanded study on lower grade glioma, more statistically significant genes were found. There were reports with findings consistent with those of ours on hsa-miR-1262, hsa-miR-455-3p, *LPAL2* [83,84,85,86], and some reports even contradicted our findings on hsa-miR-204-5p and hsa-miR-197-3p [87,88]. Although related studies are still insufficient, the previous report about hsa-miR-186-5p in glioblastoma contradicted our findings [89]. Interestingly, the survival pattern of *LINC01551* and *INHBA-AS1* related to this miRNA also were different in astrocytoma (Figure 5C and Appendix A). This compelled us to think about the possibility of producing different results for each cancer cell type, even subtypes. In our study, this means that the expression of the same gene may have different biological effects as per the cancer subtype.

## 5. Conclusions

Genes participating in the ceRNA network in OD can provide different tumor microenvironments via potassium ion channels and composition changes in ECM, and these may cause differences in survival due to the 1p/19q codeletion. Our study also demonstrates that subtypes of glioma have a high *IDH* mutation frequency and that these are known to improve survival and exhibit similar traits; however, different effects can be derived from the same gene, and this would result in survival differences in different subtypes such as ODs and astrocytomas. However, our study is based on in silico analysis with limited knowledge about the in vivo expressions and functions of many lncRNAs against endogenous mRNA. Thus, further vigorous studies are required to validate this finding with in vivo and in vitro experiments in the near future.

## Figures and Tables

**Figure 1 cancers-12-02543-f001:**
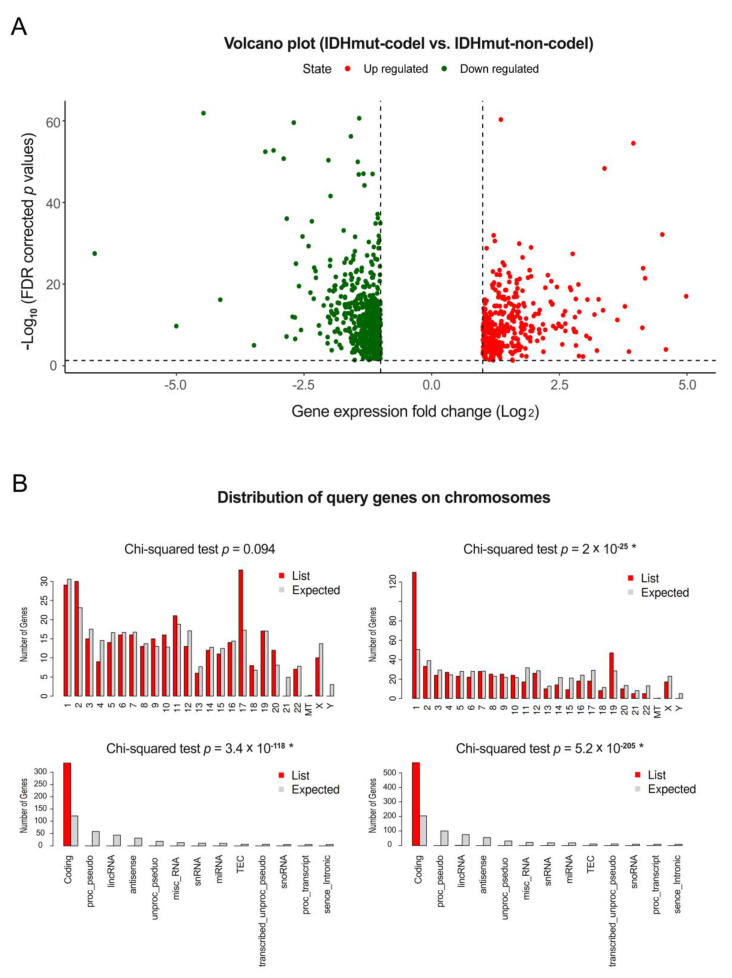
Volcano plots and distribution of genes on chromosomes for DEcodingRNAs between the OD 1p/19q codeletion group and the non-codeletion group. The volcano plot of DEcodingRNAs. The red dots represent significantly upregulated genes, the green dots represent significantly downregulated genes (|log2 FC| ≥ 1 and FDR < 0.05) (**A**). Histogram of input gene sets and distribution on chromosomes for upregulated DEcodingRNAs (left) and downregulated DEcodingRNAs (right) (**B**). * indicates significant *p* value.

**Figure 2 cancers-12-02543-f002:**
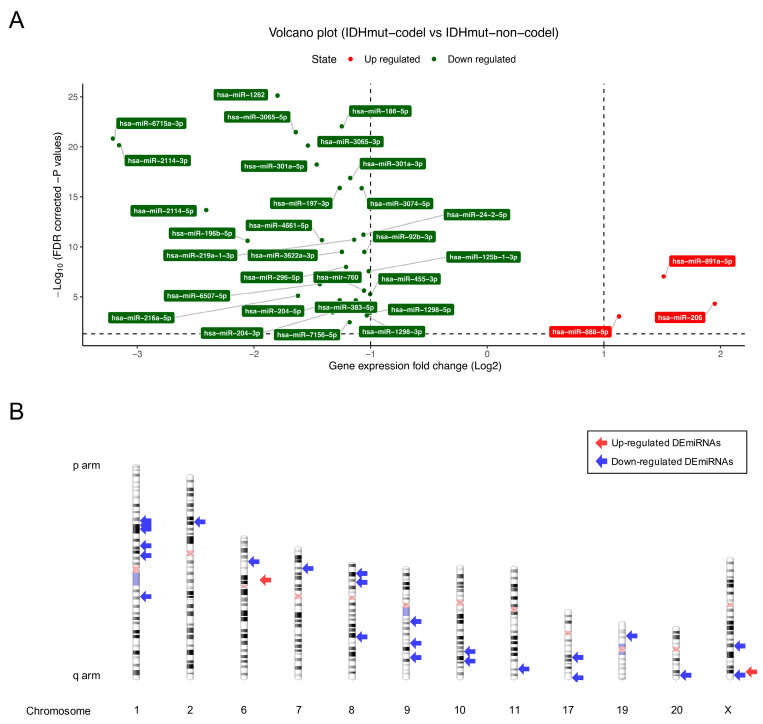
Volcano plot and distribution of genes on chromosomes for DEmiRNAs between OD 1p/19q codeletion group and non-codeletion group. The volcano plot of DEmiRNAs (**A**). The red dots represent significantly upregulated miRNAs in OD’s patients with 1p/19q codeletion, the green dots represent significantly downregulated miRNAs in OD’s patients with 1p/19q codeletion (|log2 FC| ≥ 1 and FDR < 0.05). The ideogram plot of DEmiRNAs (**B**). The red regions of chromosomes represent centromeres and the light blue regions represent highly condensed heterochromatin regions. Red arrows represent the distribution of upregulated DEmiRNAs and blue arrows represent the distribution of downregulated DEmiRNAs on the chromosomes.

**Figure 3 cancers-12-02543-f003:**
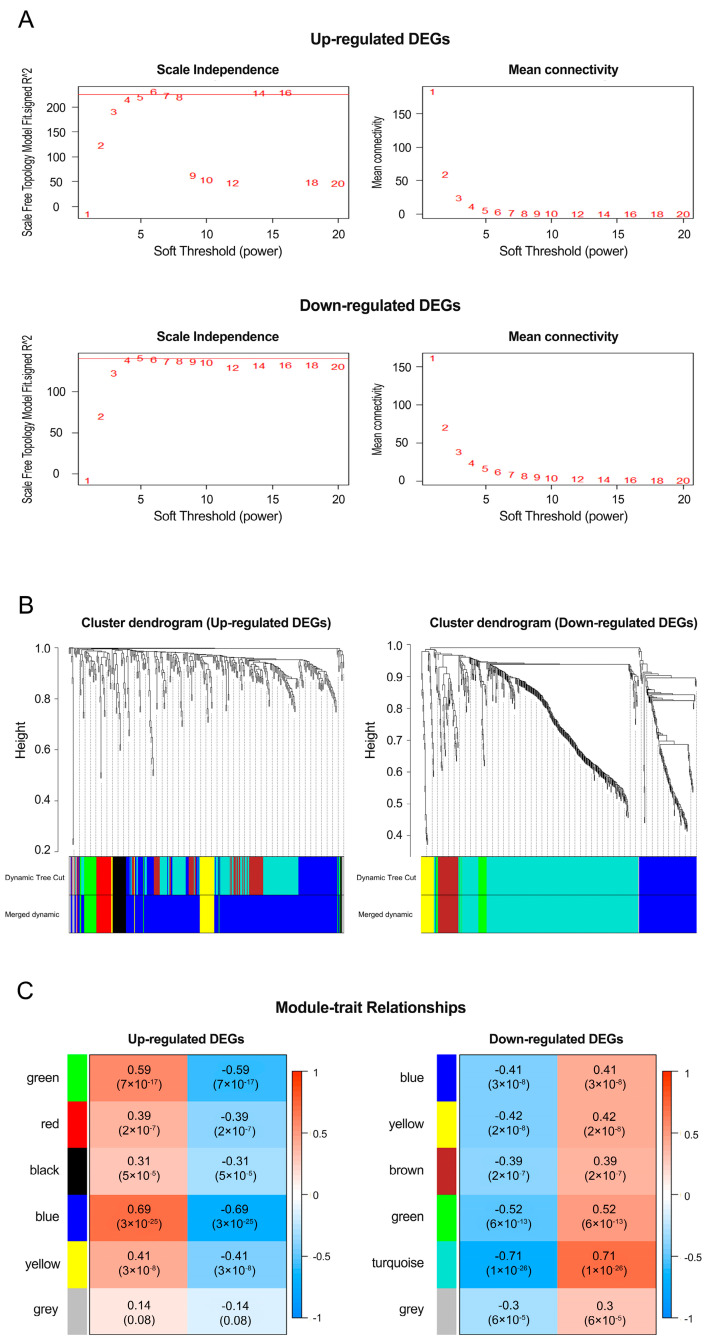
Clustering and module identification between DEcodingRNAs and DEncRNAs using WGCNA. Decision of soft-thresholding powers for scale-free topology (β) and mean connectivity (**A**). Hierarchical clustering dendrogram based on the dissimilarity between DEcodingRNAs and DEncRNAs. The first colored rows indicate modules with highly interconnected genes. The second colored rows indicate merged modules with high correlation (**B**). Module-trait relationships are calculated using correlations between module eigengene values and clinical traits. Modules were labeled using correlation coefficients and *p* values (**C**).

**Figure 4 cancers-12-02543-f004:**
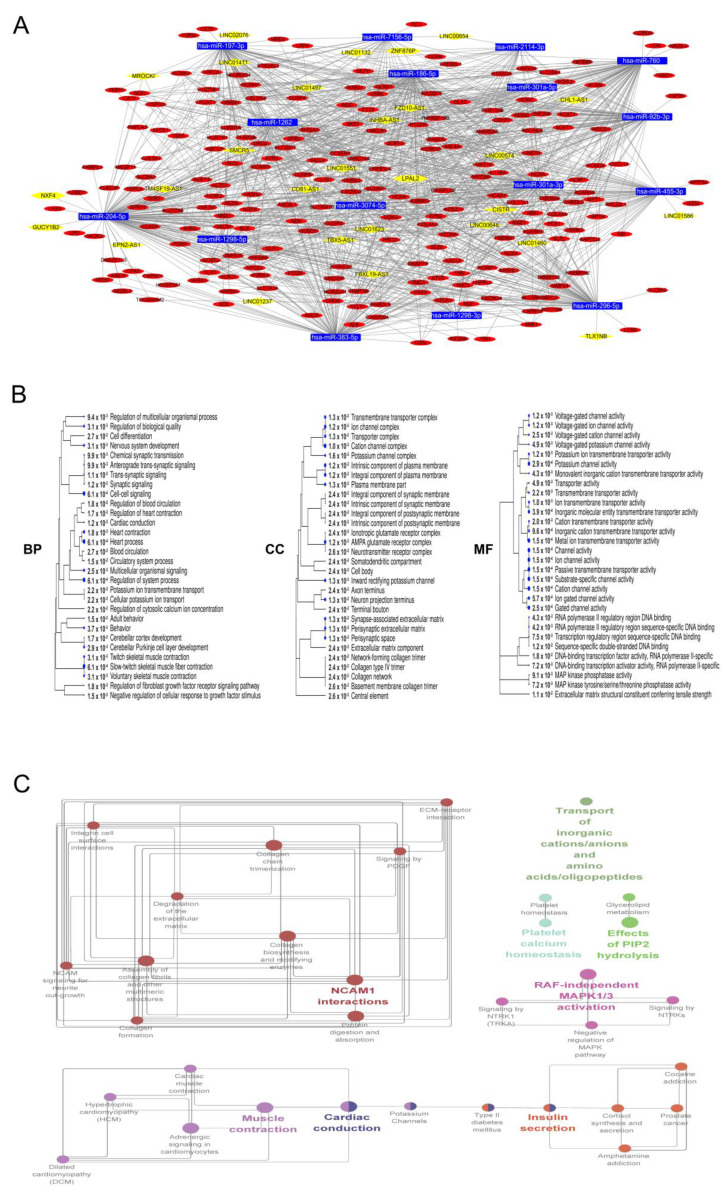
Construction and biological identification of the ceRNA network comprising upregulated DEcodingRNAs and DEncRNAs expected to be candidate targets of downregulated DEmiRNAs. The visualization of the ceRNA network. Blue rectangles indicate downregulated DEmiRNAs (**A**). Red circles and yellow rhombuses indicate upregulated DEcodingRNAs, upregulated DElncRNAs, and DEpseudogenes from the blue module. The results of GO analyses of DEcodingRNAs belonging to ceRNA network, include biological process (BP), cellular component (CC), and molecular function (MF) (**B**). Annotations of the KEGG and reactome pathway enrichment analysis (**C**).

**Figure 5 cancers-12-02543-f005:**
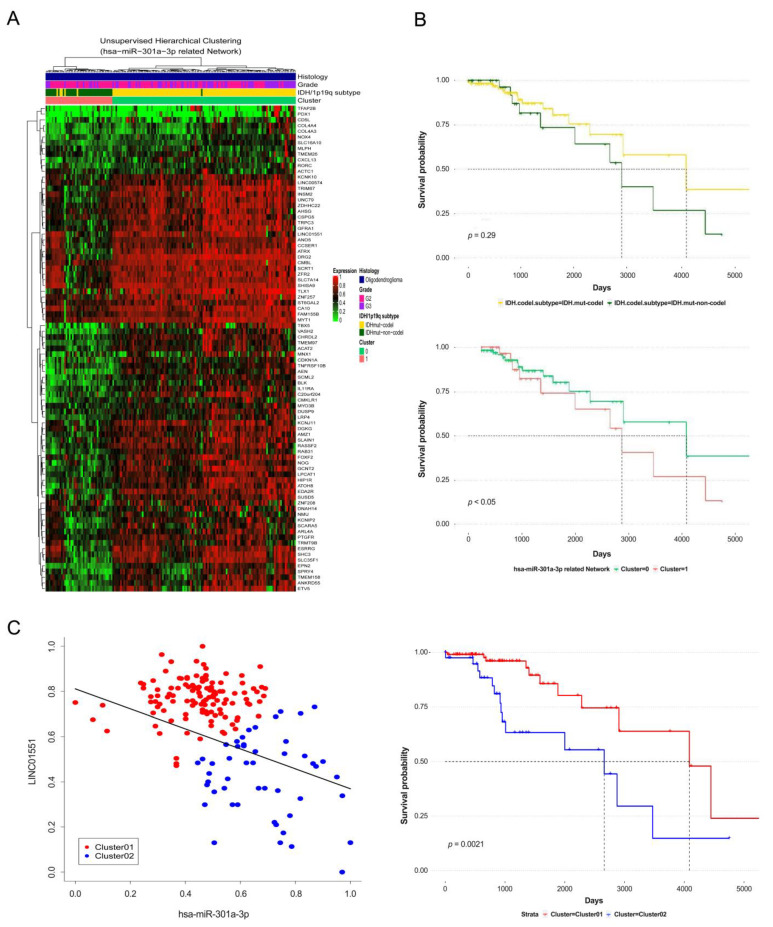
Evaluation of the ceRNA network through by unsupervised clustering and comparing survival to actual OD patients with *IDH* mutation. Heatmap for unsupervised analysis of DEcodingRNAs and DEncRNAs related to hsa-miR-301a-3p in ceRNA network (**A**). Kaplan–Meier plot for comparing the survival between actual oligodendroglioma patients with *IDH* mutation from TCGA database and unsupervised group (**B**). The yellow line indicates survival data for OD patients with 1p/19q codeletion and deep green line for OD patients with 1p/19q non-codeletion (Top). Olive green and coral pink line indicate groups from unsupervised clustering analysis with miR-301a-3p related ceRNA network (Bottom). Scatter plot for clustering group by K-means in OD patients with IDH mutation (**C**). Scatter plot resenting negative correlations between *LINC01551* expected with ceRNA and related to miRNA in ceRNA network (left). Kaplan–Meier survival plot according to the expression levels of ceRNA and miRNA (right).

**Table 1 cancers-12-02543-t001:** Comparison between OD and LGG using ARI evaluation of the ceRNA network.

Network	Source miRNAs	No. of Target Coding Genes	No. of Target Non-Coding Genes	ARI	
				OD (k * = 2)	LGG (k * = 3)
Network01	hsa-miR-296-5p	78	6	0.872	0.724
Network02	hsa-miR-455-3p	68	1	0.692	0.671
Network03	hsa-miR-760	92	1	0.896	0.761
Network04	hsa-miR-1298-5p	43	4	0.871	0.747
Network05	hsa-miR-197-3p	78	5	0.872	0.770
Network06	hsa-miR-301a-5p	25	2	0.801	0.756
Network07	hsa-miR-1262	62	2	0.801	0.497
Network08	hsa-miR-186-5p	99	9	0.921	0.806
Network09	hsa-miR-301a-3p	83	2	0.847	0.834
Network10	hsa-miR-383-5p	97	1	0.872	0.728
Network11	hsa-miR-2114-3p	30	2	0.778	0.637
Network12	hsa-miR-204-5p	138	7	0.824	0.714
Network13	hsa-miR-7156-5p	31	6	0.896	0.888
Network14	hsa-miR-92b-3p	76	4	0.800	0.594
Network15	hsa-miR-3074-5p	31	9	0.896	0.789
Network16	hsa-miR-1298-3p	30	2	0.778	0.704

K *, Number of agglomerative clusters obtained using ARI.

**Table 2 cancers-12-02543-t002:** Kaplan–Meier survival analysis with miRNA and ceRNA having negative correlation coefficients of less than −0.3

Subtype	miRNA Symbol	CeRNA Symbol	First Cluster			Second Cluster			*p*	FDR
			Median	95% CI *	No.	Median	95% CI *	No.		
OD	hsa-miR-186-5p	*INHBA-AS1*	4084	2907 to NA	113	2660	1011 to NA	48	0.002	**0.023**
	hsa-miR-301a-3p	*LINC01551*	4084	2907 to NA	115	2660	1011 to NA	46	0.002	**0.023**
	hsa-miR-186-5p	*ZNF876P*	4084	2907 to NA	102	2660	1886 to NA	59	0.023	0.168
	hsa-miR-186-5p	*TM4SF19-AS1*	4084	2907 to NA	107	2660	1886 to NA	54	0.033	0.183
	hsa-miR-186-5p	*FZD10-AS1*	4084	2907 to NA	80	2660	1886 to NA	81	0.055	0.211
	hsa-miR-204-5p	*NXF4*	4084	2907 to NA	98	2660	1886 to NA	63	0.058	0.211
	hsa-miR-301a-5p	*SMCR5*	4084	2907 to NA	99	2875	1886 to NA	62	0.113	0.356
	hsa-miR-186-5p	*LPAL2*	4084	2907 to NA	104	2660	1886 to NA	57	0.150	0.366
	hsa-miR-197-3p	*SMCR5*	4084	2907 to NA	92	2875	2000 to NA	69	0.148	0.366
	hsa-miR-197-3p	*CD81-AS1*	4084	2907 to NA	107	2660	1886 to NA	54	0.235	0.518
	hsa-miR-301a-5p	*FZD10-AS1*	2907	2875 to NA	85	2660	1886 to NA	76	0.286	0.571
	hsa-miR-186-5p	*LINC01460*	4084	2907 to NA	97	2875	2000 to NA	64	0.364	0.575
	hsa-miR-197-3p	*LINC02076*	4084	2907 to NA	98	2875	2000 to NA	63	0.366	0.575
	hsa-miR-760	*LPAL2*	3470	2907 to NA	76	2875	2000 to NA	85	0.349	0.575
	hsa-miR-204-5p	*TM4SF19-AS1*	4084	4084 to NA	81	2875	2000 to NA	80	0.410	0.602
	hsa-miR-204-5p	*EPN2-AS1*	3470	2660 to NA	86	2875	2000 to NA	75	0.472	0.615
	hsa-miR-204-5p	*SMCR5*	4084	2660 to NA	86	2875	2000 to NA	75	0.479	0.615
	hsa-miR-204-5p	*GUCY1B2*	4084	2907 to NA	101	2875	1401 to NA	60	0.503	0.615
	hsa-miR-92b-3p	*TBX5-AS1*	4084	2660 to NA	71	2907	2282 to NA	90	0.620	0.717
	hsa-miR-197-3p	*FZD10-AS1*	3470	2907 to NA	95	2875	2000 to NA	66	0.694	0.763
	hsa-miR-7156-5p	*MROCKI*	4084	1886 to NA	68	3470	2875 to NA	93	0.831	0.871
	hsa-miR-204-5p	*MROCKI*	3470	2875 to NA	89	2907	2000 to NA	72	0.933	0.933
LGG	hsa-miR-1262	*CISTR*	4445	4084 to NA	211	2000	1762 to 2875	256	0.000	**0.000**
	hsa-miR-186-5p	*INHBA-AS1*	4084	2907 to NA	213	2286	1933 to 4412	254	0.000	**0.002**
	hsa-miR-3074-5p	*INHBA-AS1*	4084	2875 to NA	254	2286	1915 to NA	213	0.000	**0.002**
	hsa-miR-186-5p	*LPAL2*	4084	2907 to NA	174	2433	1933 to 4412	293	0.002	**0.005**
	hsa-miR-186-5p	*ZNF876P*	4084	2907 to NA	178	2433	1933 to 4412	289	0.006	**0.012**
	hsa-miR-455-3p	*LINC01586*	4068	2907 to NA	251	2286	1915 to NA	216	0.005	**0.012**
	hsa-miR-186-5p	*LINC01460*	4084	2907 to NA	181	2433	1933 to 4412	286	0.014	**0.026**
	hsa-miR-186-5p	*TM4SF19-AS1*	4084	2907 to NA	186	2433	1933 to 4412	281	0.020	**0.032**
	hsa-miR-204-5p	*GUCY1B2*	3470	1891 to NA	230	2907	2286 to NA	237	0.023	**0.033**
	hsa-miR-301a-3p	*LINC01551*	4084	3470 to NA	249	2235	1915 to 3978	218	0.029	**0.037**
	hsa-miR-204-5p	*NXF4*	4084	2907 to NA	214	2433	1933 to 4068	253	0.072	0.085
	hsa-miR-301a-5p	*FZD10-AS1*	3470	2875 to NA	216	2433	1933 to NA	251	0.203	0.220
	hsa-miR-197-3p	*CD81-AS1*	4068	2907 to NA	202	2660	1933 to 4445	265	0.399	0.399

CI *, Confidence interval; Bold, Significant *p*-value with FDR, Correlation coefficients of all results were less than −0.3.

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
