# Peer review of "CeRNA Network Analysis Representing Characteristics of Different Tumor Environments Based on 1p/19q Codeletion in Oligodendrogliomas"

_cancers, 2020, doi:10.3390/cancers12092543_

Round 1
Reviewer 1 Report
The hypothesis is presented that modules of DE coding or noncoding genes may correlate with the 1p/19q deletion in different subtypes of Oligodendrogliomas (OD). The database TCGA provides a wealth of samples (168) with 127 exhibiting the 1p/19q deletion. The drill down of over-expressed (chr. 17) and underexpressed (chr. 1 and 19) in 1p/19q non-deletion or deletion groups appears to be consistent with the concept of a whole deletion of chromosomes 1p and 19q. A problem arises, however, in the interpretation of the data, particularly the miRNA network (Table 1). With the examination of the source miRNAs, there appears to be a significant compilation of miRNAs and networks from chromosome 1 (ie. miR-1262, miR-197, miR-760, miR-186-5p, miR-7156, miR-92b), a striking number of miRNAs from chromosome 9 (miR-465, miR-204, miR-3074) and a surprising absence of miRNAs from chromosome 19q. Given the hypothesis, I would have expected some significant discussion of the correlation between miRNAs that should have been deleted (haploinsufficiency) and the negative correlation coefficients. In other words, why is there a plethora of Chromosome 1p based miRNAs (and chromosome 9) and an absence of chromosome 19q. Is this consistent with the hypothesis?
Minor points- Figures S2, 3 and 4 are hard to resolve even when amplified. Larger font sizes are needed
Author Response
Reviewer #1
The hypothesis is presented that modules of DE coding or noncoding genes may correlate with the 1p/19q deletion in different subtypes of Oligodendrogliomas (OD).
The database TCGA provides a wealth of samples (168) with 127 exhibiting the 1p/19q deletion. The drill down of over-expressed (chr. 17) and underexpressed (chr. 1 and 19) in 1p/19q non-deletion or deletion groups appears to be consistent with the concept of a whole deletion of chromosomes 1p and 19q.
Point 1:
A problem arises, however, in the interpretation of the data, particularly the miRNA network (Table 1). With the examination of the source miRNAs, there appears to be a significant compilation of miRNAs and networks from chromosome 1 (ie. miR-1262, miR-197, miR-760, miR-186-5p, miR-7156, miR-92b), a striking number of miRNAs from chromosome 9 (miR-465, miR-204, miR-3074) and a surprising absence of miRNAs from chromosome 19q.
Given the hypothesis, I would have expected some significant discussion of the correlation between miRNAs that should have been deleted (haploinsufficiency) and the negative correlation coefficients. In other words, why is there a plethora of Chromosome 1p based miRNAs (and chromosome 9) and an absence of chromosome 19q. Is this consistent with the hypothesis?
Response1:
In this study, we primarily hypothesised two patterns of ceRNA networks in the oligodendroglioma (OD) patients with IDH mutation. The first hypothesis was that there might be the network of up-regulated mRNAs, lncRNAs and down-regulated miRNAs in the OD tumor systems. The second hypothesis was opposite of the first that there might be the network of down-regulated mRNAs, lncRNAs and up-regulated miRNAs in OD. Based on these concepts, we performed differential expression analyses between 1p/19q codeleted patients and 1p/19q non-codeleted patients. During this process, we found a differentially expressed miRNA on chromosome 19 (hsa-miR-24-2-5p). However, we discovered the genes with differential expression as well as the network of sharing MRE site with DEmRNAs, DElncRNAs and DEmiRNAs. According to our result, there were many DEmiRNAs on chomosome 1 that were satisfied with both above conditions. Yet, DEmiRNA located on chromosome 19 did not meet the two conditions. Moreover, we filtered out the lowly expressed DEmiRNAs by requiring an overall average expression RPM (reads per million) ≥ 3 (We added this specific information and highlighted with red color in the manuscript Line 100~102 and legend of Figure S2A).
We believe that many miRNAs located on chromosome 19 were excluded during this process. Resultantly, although there was a difference in expression on chromosome 19, they do not seem be included in the network (Please refer to table 1). Moreover, we have attached the ideogram plot of DEmiRNAs (Figure S2B) in order to aid the readers for better understanding of this study (We added this specific information and highlighted with red color in the manuscript Line 117~118 and legend of Figure S2B). Furthermore, we were also interested in the reviewer's comments about discovery of a plethora of down regulated miRNAs based on chromosome 9. We found reports that 1p/19qcodeleted high grade oligodendrogliomas is associated with 9p loss [1,2]. In this study, we focused on problem between 1p/19q codeletion and non-codeletion. However, we think that this phenomenon could importantly affect the prognosis in oligodendroglioma patient with 1p/19q-codeletion so we would like to do further research about this issue as our next project. We are indeed thankful for the reviewer's valuable comments.
The ideogram plot of DEmiRNAs was added to supplementary figure S2B.
Point 2:
Figures S2, 3 and 4 are hard to resolve even when amplified. Larger font sizes are needed.
Respones 2:
As the reviewer has pointed out, we carefully revised and upgraded the resolution of the Figures S2, 3, 4 and Figure 4.
Figure S2
Before correction
After correction
Figure S3
Before correction
After correction
Figure S4
Before correction
After correction
Figure 4
We added Y-axis values in Figure 4B.
Before correction
After correction
figures please see the attachment.
[References]
- Reyes-Botero, G.; Dehais, C.; Idbaih, A.; Martin-Duverneuil, N.; Lahutte, M.; Carpentier, C.; Letouze, E.; Chinot, O.; Loiseau, H.; Honnorat, J., et al. Contrast enhancement in 1p/19q-codeleted anaplastic oligodendrogliomas is associated with 9p loss, genomic instability, and angiogenic gene expression. Neuro Oncol 2014, 16, 662-670, doi:10.1093/neuonc/not235.
- Michaud, K.; de Tayrac, M.; D'Astous, M.; Duval, C.; Paquet, C.; Samassekou, O.; Gould, P.V.; Saikali, S. Contribution of 1p, 19q, 9p and 10q Automated Analysis by FISH to the Diagnosis and Prognosis of Oligodendroglial Tumors According to WHO 2016 Guidelines. PLoS One 2016, 11, e0168728, doi:10.1371/journal.pone.0168728.

Reviewer 2 Report
The work by Ahn and co authors is well written and of potential interest. There are some issues that should be addressed before assessing its real potential.
- The introduction is lacking somehow of focus. The general aim of the work can be bradly understood but it needs to be described more specifically. In addition, ceRNAs, which are the focus of the article even from the title, are just nominated twice and without proper introduction and description.
- The methods are lacking too in their core. Since the journal accepts theoretical results, there is no overt concern about the lack of any possible and minimal wet-lab-based validation, but the starting datasets should be clearly indicated for reproducibility. Indeed, it would be interesting to know how many datasets have been merged, the source and the kind of platform used. Importantly, it is important to know how many and for which RNA-species the data are the result of RNA-seq or of array-based experiments. Thus, it is mandatory that the authors describe exactly the datasets and experiments that were analyzed, disclosing every detail allowing full reproducibility.
- Even if it could be considered a minor point, the authors should modify their discussion with the introduction of a limitations paragraph. Indeed, while prediction softwares are growing in number, their use is based on very limited kwowledge about the real expression and function of many ncRNAs. This is particularly true for lncRNAs which have a very complex and changing behaviour in different situations.
Author Response
Reviewer #2
The work by Ahn and coauthors is well written and of potential interest. There are some issues that should be addressed before assessing its real potential.
Points 1:
The introduction is lacking somehow of focus. The general aim of the work can be bradly understood but it needs to be described more specifically. In addition, ceRNAs, which are the focus of the article even from the title, are just nominated twice and without proper introduction and description.
Response 1:
As the reviewer has mentioned, we have re-written the introduction with additional information in order to focus more specifically. More details about the research on ceRNA were added. We highlighted these modified contents in red color in the manuscript (Line 55~75 and 76~77).
Points 2:
The methods are lacking too in their core. Since the journal accepts theoretical results, there is no overt concern about the lack of any possible and minimal wet-lab-based validation, but the starting datasets should be clearly indicated for reproducibility.
Indeed, it would be interesting to know how many datasets have been merged, the source and the kind of platform used.
Importantly, it is important to know how many and for which RNA-species the data are the result of RNA-seq or of array-based experiments. Thus, it is mandatory that the authors describe exactly the datasets and experiments that were analyzed, disclosing every detail allowing full reproducibility.
Response 2:
We have now supplied more detailed methodological information in the methods section for allowing reproducibility. We highlighted these contents in red color in the manuscript (Line 83~104).
Points 3:
Even if it could be considered a minor point, the authors should modify their discussion with the introduction of a limitations paragraph. Indeed, while prediction softwares are growing in number, their use is based on very limited kwowledge about the real expression and function of many ncRNAs. This is particularly true for lncRNAs which have a very complex and changing behaviour in different situations.
Response 3:
As the reviewer has pointed out, we have added limited points of our research in the discussion session. We deeply appreciate the reviewer's valuable comments and we highlighted these amendments in red color in the manuscript accordingly (Line 434~437).
Round 2
Reviewer 1 Report
The addition of Figure S2b is a great addition. However, it is difficult to interpret which arrows refer to which chromosomes (and what does blue on the chromosomes mean). I would suggest that the importance of this figure is such that it should be moved into the manuscript (not supplementary) and the distribution of genes on the chromosomes DEmiRNAs better delineated.
Author Response
Point 1:
However, it is difficult to interpret which arrows refer to which chromosomes (and what does blue on the chromosomes mean).
Response1:
As the reviewer has mentioned, we have explained the ideogram in more detail by words in order to aid the readers for better understanding. We highlighted these contents in red color in the manuscript (Line 120~121, 187, 196~203 and legend of Figure 2B).
Point 2:
I would suggest that the importance of this figure is such that it should be moved into the manuscript (not supplementary) and the distribution of genes on the chromosomes DEmiRNAs better delineated.
Response 2:
We have only displayed chromosomes that correspond to the results of DEmiRNAs for a better delineation of gene distribution and changed supplementary Figure S2 to Figure 2.
Additional amendments:
Additional amendments were made and highlighted these contents in red color in the manuscript (Line 87~89).
Before correction
We downloaded harmonized data of the reference of hg38 (platform “Illumina HiSeq") for 189 patients with OD who had IDH mutation information from TCGA database.
After correction
We downloaded harmonized data from TCGA database as reference of hg38 (platform “Illumina HiSeq") for 189 patients with OD who had IDH mutation information and selected 168 patients with IDH mutation.

Reviewer 2 Report
I think the authors provided all answers to my requests
Author Response
We would like to thank the reviewers for their valuable comments on our paper.
We provided all answers to reviewer's comments.